# Seroepidemiological Survey of Chronic Chagas Disease in a Rural Community in Southern Bahia, Brazil, Using Recombinant Chimeric Antigens

**DOI:** 10.3390/pathogens12101222

**Published:** 2023-10-07

**Authors:** Neci Matos Soares, Nilo Manoel Pereira Vieira Barreto, Marina Morena Brito Farias, Cíntia de Lima Oliveira, Weslei Almeida Costa Araújo, Joelma Nascimento de Souza, Márcia Cristina Aquino Teixeira, Noilson Lázaro Sousa Gonçalves, Daniel Dias Sampaio, Tycha Bianca Sabaini Pavan, Paola Alejandra Fiorani Celedon, Nilson Ivo Tonin Zanchin, Fred Luciano Neves Santos

**Affiliations:** 1Department of Clinical and Toxicological Analysis, Faculty of Pharmaceutical Sciences, Federal University of Bahia, Salvador 40170-115, Bahia, Brazil; neci@ufba.br (N.M.S.); nilomanoel@gmail.com (N.M.P.V.B.); marifariias@gmail.com (M.M.B.F.); cintianeue@outlook.com (C.d.L.O.); wesleicosta11@hotmail.com (W.A.C.A.); joelmandesouza@gmail.com (J.N.d.S.); marciat@ufba.br (M.C.A.T.); 2Advanced Public Health Laboratory, Gonçalo Moniz Institute, Oswaldo Cruz Foundation (Fiocruz-BA), Salvador 40296-710, Bahia, Brazil; noilson.goncalves@fiocruz.br (N.L.S.G.); tycha.pavan@fiocruz.br (T.B.S.P.); 3Brazil’s Family Health Strategy, Municipal Health Department, Tremedal City Hall, Tremedal 45170-000, Bahia, Brazil; diassampaio@gmail.com; 4Molecular Biology of Trypanosomatids Laboratory, Carlos Chagas Institute, Oswaldo Cruz Foundation (Fiocruz-PR), Curitiba 81310-020, Paraná, Brazil; paola.fiorani@fiocruz.br; 5Structural Biology and Protein Engineering Laboratory, Carlos Chagas Institute, Oswaldo Cruz Foundation (Fiocruz-PR), Curitiba 81310-020, Paraná, Brazil; nilson.zanchin@fiocruz.br; 6Integrated Translational Program in Chagas Disease from Fiocruz (Fio-Chagas), Oswaldo Cruz Foundation (Fiocruz-RJ), Rio de Janeiro 21040-360, Rio de Janeiro, Brazil

**Keywords:** chronic chagas disease, screening, serologic diagnosis, prevalence, seroepidemiology, active case finding, epidemiological surveillance

## Abstract

Chagas disease (CD), caused by the parasite *Trypanosoma cruzi*, is a neglected tropical disease with life-threatening implications. In this study, we conducted a seroepidemiological survey to determine the prevalence and clinical profiles of CD in 217 individuals from an impoverished rural community in Southern Bahia, Brazil. The overall prevalence of CD in the studied community was 0.92%, detected through latent class analysis (LCA). Two individuals tested positive for anti-*T. cruzi* IgG, both being male farmers. One case was a 22-year-old man born in Camamu, with no evidence of congenital transmission, suggesting other routes of transmission such as vector-borne transmission due to migratory activities. The other case was a 69-year-old man born in São Felipe, who had lived in an adobe/brick house and had a pacemaker due to cardiac involvement caused by CD. The prevalence in this community was lower than expected, given the socioeconomic conditions and environmental factors that contribute to *T. cruzi* transmission. This could be attributed to the implementation of preventive measures and vector control programs by the Brazilian Government. However, continuous monitoring and surveillance are essential to sustain control efforts and detect any potential re-emergence of the disease. While the overall prevalence was low, the detection of positive cases underscores the need for continued surveillance and control measures in vulnerable populations, such as rural communities. Active surveillance, early diagnosis, and timely treatment are crucial in preventing disease progression and complications, thereby enhancing the effectiveness of screening and treatment programs.

## 1. Introduction

Chagas disease (CD) is a life-threatening neglected tropical disease caused by the parasite *Trypanosoma cruzi* [1]. Transmission primarily occurs through infected hematophagous insects, commonly known as kissing bugs [2,3]. Additionally, *T. cruzi* can be transmitted through blood transfusion, organ transplantation, consumption of contaminated food or drinks, and from mother to child during pregnancy [4]. In the context of the classic route of transmission, the disease is associated with the living conditions of the population, including risk factors such as residing in mud houses with cracks, having a chicken coop in close proximity to the residence, keeping domestic animals in the peridomicile, and living near forested areas [2].

The disease presents in two phases: acute and chronic. During the acute phase, which typically lasts a few weeks or months, individuals may experience symptoms such as fever, fatigue, body aches, and localized swelling at the site of the insect bite. However, most infected individuals either remain asymptomatic or have mild symptoms, posing challenges for diagnosis [5]. If left untreated, the infection can progress to the chronic phase, which can persist for several years or even decades. In this phase, the parasite can cause severe damage to the heart, digestive system, and other organs [6]. Cardiac complications, including cardiomyopathy, arrhythmias, and heart failure, are the leading causes of morbidity and mortality in CD [7,8].

Chagas disease is endemic in 21 Latin American countries, where an estimated 6–7 million people are infected, resulting in approximately 7500 deaths annually [9,10]. It is also becoming a global health concern due to increased migration and travel from endemic to non-endemic regions. Cases have been reported in non-endemic areas such as the United States, Canada, Europe, Asia, and Oceania [11,12]. In Brazil, the country where *T. cruzi* was first identified [1], CD is a significant public health problem. Indeed, the World Health Organization estimates that 1.1 million people are infected with *T. cruzi*, and 25.4 million people are at risk of infection [9]. However, the actual number of chronically infected individuals remains unknown, despite progress in interrupting CD transmission in Brazil. Consequently, the disease’s status as a health concern in Brazil persists.

The disease imposes substantial social and economic burdens on affected individuals, families, and healthcare systems, particularly affecting vulnerable populations in impoverished rural areas with limited access to healthcare. Efforts are underway to improve access to diagnosis, treatment, and care for CD. Active surveillance, early diagnosis, and timely treatment are crucial in preventing disease progression and its complications. Therefore, the use of newer CD screening technology is crucial for evaluating at-risk Brazilian populations, potentially revealing differences from older screening studies. Recognizing the challenges posed by CD, our study aimed to investigate the prevalence of chronic CD among individuals living in a rural community in Southern Bahia, representing an important step in understanding the disease burden in vulnerable populations.

## 2. Materials and Methods

### 2.1. Study Area

The study was conducted in the Zumbi dos Palmares Settlement (ZPS; 14°01′45 S/39°10′46 W, Figure 1), a rural community located 12 km from the municipality of Camamu in the southwestern region of Bahia, approximately 195 km from Salvador, the capital of Bahia [13]. ZPS is situated on the former Brahma Farm and covers an area of about 400 hectares, housing 251 individuals from 50 families. The majority of residents have family ties, and many receive government assistance, such as *Bolsa Família* program, to supplement their income. Seasonal migration among residents to other states occurs during the sugarcane and garlic harvests as an alternative means of increasing family income. ZPS exhibits environmental characteristics, including sandy soil enriched with organic matter and a hot, humid climate. The region experiences a prolonged summer and a brief winter. Temperatures typically range from 20 °C to 31 °C throughout the year, and rainfall is evenly distributed. April records the highest average rainfall at 99 mm, while September is the least rainy month, averaging 46 mm of precipitation [14].

### 2.2. Study Design and Population

The population included in this cross-sectional seroepidemiological study consisted of individuals of all genders and ages residing in ZPS. All ZPS residents were invited to participate, and only those who provided consent had their blood collected for the study. No other inclusion or exclusion criteria were applied. In addition to assessing the serological status of Chagas disease, other variables, such as age group, and years of education, were analyzed.

### 2.3. Serology Testing

We performed anti-*T. cruzi* immunoassays using ELISA with four *T. cruzi* recombinant chimeric antigens: IBMP-8.1, IBMP-8.2, IBMP-8.3, and IBMP-8.4. The synthesis of these antigens followed established protocols [15]. Synthetic genes were obtained from GenScript (Piscataway, NJ, USA) and subcloned into the pET28a vector. *Escherichia coli*-Star (DE3) cells were employed for antigen expression and cultured in Luria-Bertani medium supplemented with 0.5 mM IPTG (isopropyl-β-D-1-thiogalactopyranoside). His-tagged chimeric antigens were purified using ion exchange and affinity chromatography, followed by quantification with fluorometry (Qubit 2.0, Invitrogen Technologies, Carlsbad, CA, USA) following the manufacturer’s instructions.

For detecting anti-*T. cruzi* antibodies, we utilized the IBMP antigens based on prior studies [16,17,18,19]. The assays were performed on transparent 96-well flat-bottom microplates (Nunc, Roskilde, Denmark) coated with one of the chimeric IBMP antigens. The coating concentrations were 12.5 ng for IBMP-8.2 and 25 ng for IBMP-8.1, IBMP-8.3, and IBMP-8.4 per well, using a coating buffer (0.05 M carbonate bicarbonate, pH 9.6). Coating and blocking were simultaneously carried out using a synthetic buffer (WellChampion; Kem-En-Tec Diagnostics A/S, Taastrup, Denmark) according to the manufacturer’s instructions. Serum samples, diluted 1:100 in 0.05 M phosphate-buffered saline (PBS; pH 7.4), were added to the coated wells, and the microplates were incubated at 37 °C for 60 min. Subsequently, the wells were washed with PBS-0.05% Tween-20 (PBS-T; pH 7.4) to remove non-adsorbed material, followed by another incubation at 37 °C for 30 min with 100 μL of HRP-conjugated goat anti-human IgG (Bio-Manguinhos, FIOCRUZ, Rio de Janeiro, Brazil) diluted 1:40,000 in PBS. Following another wash cycle, 100 μL of TBM substrate (Kem-En-Tec Diagnostics A/S, Taastrup, Denmark) was added to the wells to detect the formation of immune complexes. Incubation was conducted for 10 min at room temperature in the dark. Colorimetric reactions were stopped by adding 50 μL of 0.3 M H_2_SO_4_ to each well. Optical density was measured using a SPECTRAmax 340PC microplate reader with a 450 nm filter (Molecular Devices, San Jose, CA, USA), and background values were subtracted from the measurements.

### 2.4. Data Analysis

Patient characteristics and results were recorded in an encrypted electronic data collection database using SPSS software v.19.0 for Windows (SPSS Inc, Chicago, IL, USA). Categorical variables were described using frequencies and percentages, while continuous variables were described using means, standard deviations, and ranges. Prism software (version 10; GraphPad, San Diego, CA, USA) was utilized for analyzing and visualizing ELISA data. To establish relevant cutoff values (CO) for IBMP-ELISA, ten *T. cruzi*-reactive and ten *T. cruzi*-non-reactive samples were assessed simultaneously in all microplates. These samples were previously characterized as positive or negative based on two serological tests following international guidelines [20,21]. The CO values were determined by calculating the largest area under the ROC curve, which defined the maximum optical density (OD) required to discriminate between reactive and nonreactive *T. cruzi* samples. The results were expressed as a reactivity index (RI), representing the ratio of sample OD to CO. Samples with RI values > 1.00 were considered positive, while samples with RI values falling within the indeterminate zone (RI values of 1.0 ± 10%) were classified as inconclusive.

In the absence of a gold standard for diagnosing chronic CD, latent class analysis (LCA) was employed as a statistical approach for serological classification of *T. cruzi* infection. LCA is a well-established and validated multivariate statistical approach based on categorical indicators or latent variables [19,22,23]. Four indicators representing IBMP-8.1, IBMP-8.2, IBMP-8.3, and IBMP-8.4 were defined to characterize the latent variable for diagnosing *T. cruzi* infection. The latent class response patterns classified a sample as *T. cruzi* reactive if it showed positive results in at least two different chimera-based assays (a posteriori probability ranged from 87.9 to 100%). Conversely, a sample was considered non-reactive for *T. cruzi* if all four chimeric antigens gave a non-reactive result or if only one of the antigens was positive (a posteriori probability ranged from 0 to 0.8%). Sixteen response patterns were identified and divided into five categories (P1 to P5). To provide a visual representation of the classification process, we constructed a flow chart, which is presented in Figure 2.

## 3. Results

A total of 217 individuals (86.5%; 217/251) participated in this study. The median age was 27 years (interquartile range [IQR] 13.3–49), with a higher proportion of women (51.4%) than men (48.9%) participating. Data for five participants could not be retrieved. The age distribution revealed that the majority of participants were below 40 years old (67%), while approximately 8% were aged between 41 and 50, 11% were aged between 51 and 60, and 14% were older than 60 years. Regarding education level, the majority of participants (77.4%) had up to eight years of formal schooling, 21.7% were illiterate and only 2% attended university without completing their studies. Out of the 212 participants with available income information, approximately 50% reported an annual personal income of USD 600 and, the other 50% reported a mean of USD 3200.

Among the 217 serum samples analyzed, latent class analysis detected anti-*T. cruzi* IgG in two samples, resulting in an overall prevalence of 0.92% (2/217). Both samples were classified as P5, indicating positivity for all four IBMP proteins with a posteriori probability of 100%.

These two individuals tested positive for anti-*T. cruzi* IgG were male farmers. One sample belonged to a 22-year-old man born in Camamu, whose family members, including his mother, tested negative for Chagas disease. He migrated to other cities in the states of Bahia, Minas Gerais, or Espírito Santo once or twice per year during the sugarcane and garlic harvests. The other sample was obtained from a 69-year-old man born in São Felipe, an endemic area for Chagas disease, who reported living in an adobe/brick house during his childhood and being constantly exposed to kissing bug bites. Unlike the younger man, he was already aware of his clinical condition and required a pacemaker due to cardiac involvement caused by Chagas disease. The remaining 215 samples were classified as *T. cruzi*-negative, with all samples showing a negative result for all four IBMP proteins (classified as P1).

## 4. Discussion

Chagas disease (CD) continues to be a significant public health issue in Brazil, with a high burden of undiagnosed cases [21]. In this study, we conducted a seroepidemiological survey to determine the prevalence of CD in an impoverished rural community in Southern Bahia. While the study site falls under a low endemicity category for CD, it is endemic for other infections, including HTLV, schistosomiasis, and intestinal parasites, with strongyloidiasis being notably prevalent. The evaluation of CD positivity is part of a broader project focused on diagnosing and treating individuals for multiple infections. Our findings highlight the importance of active surveillance and early detection of CD in vulnerable populations. The overall prevalence of chronic CD in the studied community was 0.92%, as detected by latent class analysis (LCA). This prevalence is relatively low compared to the estimated national prevalence of CD in Brazil [24]. However, it is important to note that even in areas of low prevalence, identifying positive cases through seroepidemiological surveys is crucial for timely diagnosis and management.

The absence of positive cases among the family members of one individual infected with *T. cruzi* suggests that congenital transmission was not the cause of his infection. Other possible routes of transmission, such as vector-borne transmission, should be considered, especially considering his migratory activities during the sugarcane and garlic harvests. Although we were unable to conduct clinical follow-up due to his temporary residence in another state during our visits to the Zumbi dos Palmares settlement, he was referred to primary care in his municipality of residence. The other individual infected with *T. cruzi* born in an endemic city of Bahia, was already diagnosed with arrhythmogenic cardiomyopathy attributable to CD. This emphasizes the long-term health consequences and potential severity of the disease if left untreated.

The prevalence of CD in the studied rural community was lower than expected, considering the socioeconomic conditions and environmental factors that can contribute to *T. cruzi* transmission. Although present, we observed few mud houses with cracks in the area, and the residents reported not encountering the bugs. A systematic review and meta-analysis published in 2014 reported a mean prevalence of 4.2% for Brazil, based on data from 18 states spanning 1980 to 2011 [24]. For Bahia, the same study calculated a mean prevalence of 20.4% using two available studies. The first study was carried out in Catolândia in 1989, in which infection was detected in 11.1% of 344 individuals aged 0 to >61 years [25]. The second study was performed in 2002 in Mulungu do Morro, where 25.1% of 694 individuals tested positive for Chagas disease [26]. The implementation of preventive measures and vector control programs in the region may have contributed to reducing the transmission risk. However, continuous monitoring and surveillance are necessary to ensure sustained control and detect any potential re-emergence of the disease.

The seroepidemiological survey provided valuable insights into the prevalence and clinical profiles of CD in the studied community. The survey identified two positive cases, one of which involved an individual already aware of their clinical condition. However, the other case would have remained undiagnosed without the survey, emphasizing the importance of active case finding and early detection. Prompt inclusion of this individual in primary health care ensured appropriate medical follow-up and comprehensive care.

## 5. Conclusions

In conclusion, this study sheds light on the seroepidemiological profile of CD in a rural community in Southern Bahia. Although the overall prevalence was relatively low, the detection of positive cases highlights the need for continued surveillance and control measures in vulnerable populations. Active surveillance, early diagnosis, and timely treatment remain critical in preventing disease progression and its complications [27,28,29,30,31]. Further studies are warranted to understand the dynamics of CD transmission in the region and to develop targeted interventions for effective control and elimination of the disease.

## Figures and Tables

**Figure 1 pathogens-12-01222-f001:**
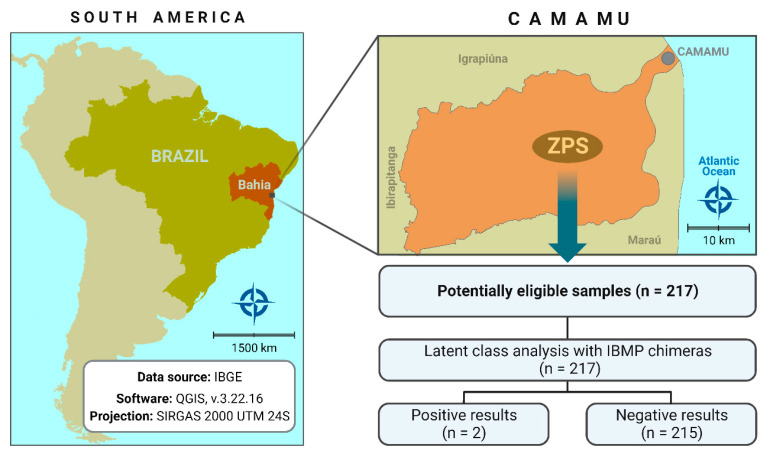
Geographic location of sample collection sites in this study. Public domain digital map was freely obtained from the Brazilian Institute of Geography and Statistics (IBGE) cartographic database in shapefile format (.shp), which was subsequently reformatted and analyzed using QGIS version 3.22.16 (Geographic Information System, Open-Source Geospatial Foundation Project. http://qgis.osgeo.org, accessed on 27 June 2023).

**Figure 2 pathogens-12-01222-f002:**
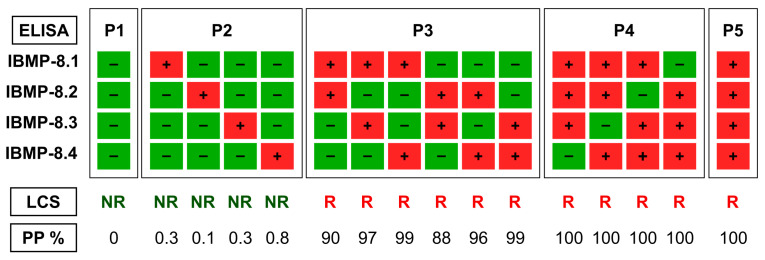
Reaction patterns of chimeric antigens in latent class analysis (LCA) used in anti-*T. cruzi* ELISA tests. LCS, latent class status; NR, nonreactive; PP, a posteriori probability; R, reactive; P1, P2, P3, P4, and P5, reaction response. A red square with a + sign denotes a positive result in ELISA-IBMP, while a green square with a − sign indicates a negative result in ELISA-IBMP.

## Data Availability

The raw data supporting the conclusions of this article are provided by the authors without reservation.

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
