# Peer review of "Seroepidemiological Survey of Chronic Chagas Disease in a Rural Community in Southern Bahia, Brazil, Using Recombinant Chimeric Antigens"

_pathogens, 2023, doi:10.3390/pathogens12101222_

Round 1
Reviewer 1 Report
The manuscript entitled “Seroepidemiological Survey of Chronic Chagas Disease in a rural Community in Southern Bahia Using Recombinant Chimeric Antigens” looks interesting and relevant. It is well written and my comments below are just suggestions for your consideration.
1.- The title of the paper indicates that they used recombinant antigens. At first, one would think that they would contrast these antigens with conventional antigens, but this was not the case. In this sense, the antigen used could be removed from the title (it is only a suggestion, it is up to you and the authors).
2. It is not clearly specified why data on skin color was obtained. In results they describe the data, but do not discuss or justify this.
3. It is not clearly specified why they chose this study site. Although the community presents unfavorable socioeconomic conditions, it is not mentioned if there is a record of the presence of the vector.
Other suggestions to the manuscript are:
Line 48. Add sexual transmission as a transmission mechanism for T. cruzi
Line 53. The chronic stage may not appear, even though the patient has not been treated. change sentence
Author Response
General Comments 1. The title of the paper indicates that they used recombinant antigens. At first, one would think that they would contrast these antigens with conventional antigens, but this was not the case. In this sense, the antigen used could be removed from the title (it is only a suggestion, it is up to you and the authors).
Reply: We appreciate the author's suggestion. Nonetheless, I would prefer to proceed with the proposed title (unless any issues arise). We are of the opinion that the progression of Chagas disease diagnosis should continue daily, and retaining the current title serves as an effort to promote the adoption of progressively advanced tools for diagnosing this significant ailment.
General Comments 2. It is not clearly specified why data on skin color was obtained. In results they describe the data, but do not discuss or justify this.
Reply: Thank you for this valuable observation. Skin color was recorded as part of our study's population characterization. However, it was not discussed in detail since Chagas disease can affect individuals of all races, ages, and genders, with certain groups, notably brown and black men in rural areas, being more exposed. The low number of positive cases in our study precluded us from establishing any significant associations.
General Comments 3. It is not clearly specified why they chose this study site. Although the community presents unfavorable socioeconomic conditions, it is not mentioned if there is a record of the presence of the vector.
Reply: This observation is of great significance. While the study site falls under a low endemicity category for Chagas disease, it is highly endemic for other illnesses, including HTLV, schistosomiasis, and various intestinal parasites, with strongyloidiasis being notably prevalent. The evaluation of Chagas disease positivity is part of a broader project focused on diagnosing and treating individuals for multiple infections. Consequently, this specific population was thoughtfully selected to provide benefits beyond Chagas disease diagnosis, encompassing the detection and treatment of other infections.
Other suggestions to the manuscript. Line 48. Add sexual transmission as a transmission mechanism for T. cruzi.
Reply: We respectfully request that the reviewer reconsider his/her criticism. While theoretically possible, sexual transmission is likely infrequent in comparison to transmission via vectors. The viability of the sexual transmission mechanism for T. cruzi lacks consensus. T. cruzi DNA may appear in the genital secretions, semen, and vaginal fluids of those infected. Instances exist where the parasite's genetic material was identified in the genital tracts of both genders. Yet, the mere presence of genetic material does not necessarily signify the presence of viable parasites capable of causing infection through sexual contact.
Other suggestions to the manuscript. Line 53. The chronic stage may not appear, even though the patient has not been treated. change sentence.
Reply: We have modified this section in the manuscript to incorporate the reviewer’s suggestions, as follows:
Before: If left untreated, the infection progresses to the chronic phase, which can persist for several years or even decades.
After: If left untreated, the infection can progress to the chronic phase, which can persist for several years or even decades.
Reviewer 2 Report
Thank you for the opportunity to review the research for submission in Pathogens, “Seroepidemiological Survey of chronic Chagas Disease in a rural community in Southern Bahia using Recombinant Chimeric Antigens”. The authors undertook to perform a serological survey of individuals with Chagas disease living in an impoverished farming area named Zombi dos Palmares Settlement in Brazil. Geographic and demographic data for this area were supplied. They screened 217 sera with a lab technique utilizing T. cruzi recombinant chimeric antigens. They assigned chronic infection status using Latent class analysis. 2 infected individuals were identified (0.92%). This was considerably lower than the anticipated prevalence of infection.
Gaining information on prevalence of Chagas in various populations will guide future screening and treatment efforts. Sharing information on serological screening practices for Chagas disease is also of interest to those involved in care of tropical diseases. Many readers of Pathogens will be interested in learning more about Chagas disease. With revision this research will be a contribution to the literature regarding this neglected disease.
My recommendations is for Major revision prior to publication:
1-Title- Add “Brazil” to the title – Outside researchers seeking epidemiological information of Chagas in Brazil will be more likely to find this paper on a literature search if Bahia is identified as Bahia Brazil .
2- Abstract /introduction – I think the abstract /introduction needs more focus on answering a specific question regarding Chagas . For instance –“To evaluate the changing incidence of Chagas in Brazil to better direct screening and treatment programs “ To apply newer Chagas screening technology to at risk Brazilians populations to see if different from older screening studies.”
3 . Introduction – I concur that any article on Chagas requires some general facts about the infection and the long-term disease effects since there is little familiarity with the disease in research and medical professionals. I recommend writing an introduction like an inverted pyramid – starting general and then focusing down to the specific question at hand. I recommend tightening up the first paragraph of the introduction about the disease in general. Include the role of living conditions. Then in the next paragraph taking the epidemiology from the world to South America and then Brazil. Then I recommend wrapping that paragraph up with the public health concerns for Brazil with Chagas. The last paragraph should make the case for why this population was chosen for screening and why the chimeric antigen test is a newer tool to apply to the screening. It is a good choice for looking for chronic Chagas with fewer false positives. It has been helpful in transfusion screening to avoid unnecessarily discarded units due to a false positive Chagas test. I think the statement about a lack of a “gold standard for diagnosing Chagas” be moved to here.
4 Methods – I need more information on why this community was selected? Are the researchers public health professionals that were asked by the community to screen? I liked the map but some of the demographics seemed less relevant (I am not sure that rainfall or the types of crops tell me much about Chagas?) . Does the area have data on housing types, indoor plumbing, electricity etc? How were the subjects recruited? Health fair? Door to door? Was permission obtained? IRB done? (If no permission or if exemption form IRB review was not obtained – I recommend Pathogens reject the paper.) Was medical follow up available to those screening positive? I recommend severely editing the part about skin tone. In my understanding, Triatomene bugs bite individuals of all skin tones and T cruzi is infectious across all ethnic types. I think the socioeconomic data reflecting housing and working conditions is more informative. If the skin tone portion is not removed, I recommend Pathogens reject the manuscript.
5 Methods – The chimeric antigen test has been applied for screening for chronic Chagas. While some basic description is helpful to the reader, likely some of the technical portion can be referenced to other papers.
6.Results – not overstated. I assume among the 200 people living in the ZPS settlement the information on the 2 infected individuals is not identifiable to them specifically or that they gave permission for the details to be published? If not, I think the details could be made more general as to avoid identification.
7 Discussion- For the discussion, emphasize the findings of the study a lower rate of infection than expected. Explain why the findings are unique to the literature – lower than prior reports of Brazilian infection. I think discussing the use of the chimeric antigen screen as more specific would be helpful here and its use in other settings. Discuss the limitations of the study – small sample, but while not randomized it was a near complete survey of the community. Remove the details about the individuals here. I think the use of the chimeric antigens deserves a paragraph on the pros and cons. this may be a reason why the incidence was lower? Requiring transportation of samples and a full laboratory to process is a negative. How is cost comparison to running 2 conventional assays? The point of the paper may be that if you go out to do a seroprevalence study on Chagas, better to use a more specific test?
Author Response
Dear Reviewer,
We extend our gratitude yiu for your insightful recommendations, which have significantly enhanced our manuscript. We believe that this revised version now offers a more refined and equitable depiction of our research. We are optimistic that the manuscript aligns well with the standards of your esteemed journal and kindly request your consideration for publication. The responses to the questions have been provided below for your reference (Manuscript ID: 1196403).
1-Title- Add “Brazil” to the title – Outside researchers seeking epidemiological information of Chagas in Brazil will be more likely to find this paper on a literature search if Bahia is identified as Bahia Brazil.
Reply: We appreciate the reviewer for bringing this deficiency to our attention. As detailed below, we have now included this information in the title:
Before: “Seroepidemiological Survey of Chronic Chagas Disease in a rural Community in Southern Bahia, Brazil, Using Recombinant Chimeric Antigens”
After: Seroepidemiological Survey of Chronic Chagas Disease in a rural Community in Southern Bahia, Brazil, Using Recombinant Chimeric Antigens
2- Abstract /introduction – I think the abstract /introduction needs more focus on answering a specific question regarding Chagas. For instance –“To evaluate the changing incidence of Chagas in Brazil to better direct screening and treatment programs “ To apply newer Chagas screening technology to at risk Brazilians populations to see if different from older screening studies.”
Reply: Thanks to the reviewer's suggestions, we have incorporated the sentences into the abstract and introduction sections as follows:
Abstract:
Before: “…While the overall prevalence was low, the detection of positive cases underscores the need for continued surveillance and control measures in vulnerable populations, such as rural communities. Active surveillance, early diagnosis, and timely treatment are crucial in preventing disease progression and complications.”
After: While the overall prevalence was low, the detection of positive cases underscores the need for continued surveillance and control measures in vulnerable populations, such as rural communities. Active surveillance, early diagnosis, and timely treatment are crucial in preventing disease progression and complications, thereby enhancing the effectiveness of screening and treatment programs.
Introduction:
Before: The disease imposes substantial social and economic burdens on affected individuals, families, and healthcare systems, particularly affecting vulnerable populations in impoverished rural areas with limited access to healthcare. Efforts are underway to improve access to diagnosis, treatment, and care for CD. Active surveillance, early diagnosis, and timely treatment are crucial in preventing disease progression and its complications. Recognizing the challenges posed by CD, our study aimed to investigate the prevalence of chronic CD among individuals living in a rural community in Southern Bahia, representing an important step in understanding the disease burden in vulnerable populations.
After: The disease imposes substantial social and economic burdens on affected individuals, families, and healthcare systems, particularly affecting vulnerable populations in impoverished rural areas with limited access to healthcare. Efforts are underway to improve access to diagnosis, treatment, and care for CD. Active surveillance, early diagnosis, and timely treatment are crucial in preventing disease progression and its complications. Therefore, the use of newer CD screening technology is crucial for evaluating at-risk Brazilian populations, potentially revealing differences from older screening studies. Recognizing the challenges posed by CD, our study aimed to investigate the prevalence of chronic CD among individuals living in a rural community in Southern Bahia, representing an important step in understanding the disease burden in vulnerable populations.
3. Introduction – I concur that any article on Chagas requires some general facts about the infection and the long-term disease effects since there is little familiarity with the disease in research and medical professionals. I recommend writing an introduction like an inverted pyramid – starting general and then focusing down to the specific question at hand. I recommend tightening up the first paragraph of the introduction about the disease in general. Include the role of living conditions. Then in the next paragraph taking the epidemiology from the world to South America and then Brazil. Then I recommend wrapping that paragraph up with the public health concerns for Brazil with Chagas. The last paragraph should make the case for why this population was chosen for screening and why the chimeric antigen test is a newer tool to apply to the screening. It is a good choice for looking for chronic Chagas with fewer false positives. It has been helpful in transfusion screening to avoid unnecessarily discarded units due to a false positive Chagas test. I think the statement about a lack of a “gold standard for diagnosing Chagas” be moved to here.
Reply: We appreciate the valuable suggestions from the reviewer. In response, we have included the following sentence in the first paragraph for clarification: “In the context of the classic route of transmission, the disease is associated with the living conditions of the population, including risk factors such as residing in mud houses with cracks, having a chicken coop in close proximity to the residence, keeping domestic animals in the peridomicile, and living near forested areas [2].”. In order to maintain a logical flow of discussion, we have divided this paragraph into two sections. Consequently, the clinical aspects of the disease have been placed in a separate second paragraph as follows:
Before: “Chagas disease (CD) is a life-threatening neglected tropical disease caused by the parasite Trypanosoma cruzi [1]. Transmission primarily occurs through infected hematophagous insects, commonly known as kissing bugs [2,3]. Additionally, T. cruzi can be transmitted through blood transfusion, organ transplantation, consumption of contaminated food or drinks, and from mother to child during pregnancy [4]. The disease presents in two phases: acute and chronic. During the acute phase, which typically lasts a few weeks or months, individuals may experience symptoms such as fever, fatigue, body aches, and localized swelling at the site of the insect bite. However, most infected individuals either remain asymptomatic or have mild symptoms, posing challenges for diagnosis [5]. If left untreated, the infection can progress to the chronic phase, which can persist for several years or even decades. In this phase, the parasite can cause severe damage to the heart, digestive system, and other organs [6]. Cardiac complications, including cardiomyopathy, arrhythmias, and heart failure, are the leading causes of morbidity and mortality in CD [7,8]..
After: “Chagas disease (CD) is a life-threatening neglected tropical disease caused by the parasite Trypanosoma cruzi [1]. Transmission primarily occurs through infected hematophagous insects, commonly known as kissing bugs [2,3]. Additionally, T. cruzi can be transmitted through blood transfusion, organ transplantation, consumption of contaminated food or drinks, and from mother to child during pregnancy [4]. In the context of the classic route of transmission, the disease is associated with the living conditions of the population, including risk factors such as residing in mud houses with cracks, having a chicken coop in close proximity to the residence, keeping domestic animals in the peridomicile, and living near forested areas [2].
The disease presents in two phases: acute and chronic. During the acute phase, which typically lasts a few weeks or months, individuals may experience symptoms such as fever, fatigue, body aches, and localized swelling at the site of the insect bite. However, most infected individuals either remain asymptomatic or have mild symptoms, posing challenges for diagnosis [5]. If left untreated, the infection can progress to the chronic phase, which can persist for several years or even decades. In this phase, the parasite can cause severe damage to the heart, digestive system, and other organs [6]. Cardiac complications, including cardiomyopathy, arrhythmias, and heart failure, are the leading causes of morbidity and mortality in CD [7,8]”.
Regarding the second paragraph, we have added the sentence at the end: " Consequently, the disease's status as a health concern in Brazil persists."
Before: Chagas disease is endemic in 21 Latin American countries, where an estimated 6-7 million people are infected and resulting in approximately 7,500 deaths annu-ally [9,10]. It is also becoming a global health concern due to increased migration and travel from endemic to non-endemic regions. Cases have been reported in non-endemic areas such as the United States, Canada, Europe, Asia, and Oceania [11,12]. In Brazil, the country where T. cruzi was first identified [1], CD is a sig-nificant public health problem. Indeed, the World Health Organization estimates that 1.1 million people are infected with T. cruzi, and 25.4 million people are at risk of infection [9]. However, the actual number of chronically infected individuals remains unknown, despite progress in interrupting CD transmission in Brazil.
After: Chagas disease is endemic in 21 Latin American countries, where an estimated 6-7 million people are infected and resulting in approximately 7,500 deaths annu-ally [9,10]. It is also becoming a global health concern due to increased migration and travel from endemic to non-endemic regions. Cases have been reported in non-endemic areas such as the United States, Canada, Europe, Asia, and Oceania [11,12]. In Brazil, the country where T. cruzi was first identified [1], CD is a sig-nificant public health problem. Indeed, the World Health Organization estimates that 1.1 million people are infected with T. cruzi, and 25.4 million people are at risk of infection [9]. However, the actual number of chronically infected individuals remains unknown, despite progress in interrupting CD transmission in Brazil. Consequently, the disease's status as a health concern in Brazil persists.
4 Methods – I need more information on why this community was selected? Are the researchers public health professionals that were asked by the community to screen? I liked the map but some of the demographics seemed less relevant (I am not sure that rainfall or the types of crops tell me much about Chagas?). Does the area have data on housing types, indoor plumbing, electricity etc? How were the subjects recruited? Health fair? Door to door? Was permission obtained? IRB done? (If no permission or if exemption form IRB review was not obtained – I recommend Pathogens reject the paper.) Was medical follow up available to those screening positive? I recommend severely editing the part about skin tone. In my understanding, Triatomene bugs bite individuals of all skin tones and T cruzi is infectious across all ethnic types. I think the socioeconomic data reflecting housing and working conditions is more informative. If the skin tone portion is not removed, I recommend Pathogens reject the manuscript.
Reply: This observation is of great significance. While the study site falls under a low endemicity category for Chagas disease, it is highly endemic for other illnesses, including HTLV, schistosomiasis, and various intestinal parasites, with strongyloidiasis being notably prevalent. The evaluation of Chagas disease positivity is part of a broader project focused on diagnosing and treating individuals for multiple infections. Consequently, this specific population was thoughtfully selected to provide benefits beyond Chagas disease diagnosis, encompassing the detection and treatment of other infections. To clarify this information, we have inserted the following sentence in the discussion:
Before: Chagas disease (CD) continues to be a significant public health issue in Brazil, with a high burden of undiagnosed cases [21]. In this study, we conducted a seroepidemiological survey to determine the prevalence of CD in an impoverished rural community in South-ern Bahia. Our findings highlight the importance of active surveillance and early detection of CD in vulnerable populations.
After: Chagas disease (CD) continues to be a significant public health issue in Brazil, with a high burden of undiagnosed cases [21]. In this study, we conducted a seroepidemiological survey to determine the prevalence of CD in an impoverished rural community in South-ern Bahia. While the study site falls under a low endemicity category for CD, it is endemic for other infections, including HTLV, schistosomiasis, and intestinal parasites, with strongyloidiasis being notably prevalent. The evaluation of CD positivity is part of a broader project focused on diagnosing and treating individuals for multiple infections. Our findings highlight the importance of active surveillance and early detection of CD in vulnerable populations.
Our investigative team comprises healthcare professionals, including physicians, pharmacists, and biochemists. To provide context about the study area, we aimed to shed light on local characteristics. Notably, the residences in this area are equipped with electricity and rely on well water for consumption. The housing is relatively simple, with few mud houses present. We have inserted the sentence in the discussion topic: “Although present, we observed few mud houses with cracks in the area, and the residents reported not encountering the bugs”
Recruitment for the study was conducted through a health campaign held at a local school. To ensure inclusivity, individuals unable to attend the designated location were visited at home, enabling widespread participation. However, we collected biological material solely from individuals who willingly participated in the study and provided informed consent by signing the consent form.
In terms of ethical considerations, our study received approval from the Institutional Review Board (IRB) for Human Research at the Gonçalo Moniz Institute, Oswaldo Cruz Foundation (FIOCRUZ), Salvador, Bahia-Brazil, under protocol number 59644422.0.0000.0040. We adhered strictly to the principles outlined in the Declaration of Helsinki, along with subsequent revisions, as well as pertinent Brazilian ethical resolutions, including Res. n° 466/1996 and n° 510/2016. Prior to participation, we obtained informed consent from all participants. To ensure confidentiality, we coded samples anonymously, safeguarding the identity of those involved.
For those testing positive, we offered medical follow-up. However, it is noteworthy that one of the infected individuals was already under the care of a medical team, while the other relocated to a different state in Brazil. We successfully established contact with the latter individual by phone and learned that he is already receiving medical supervision.
Regarding the characterization of the study population, we initially used skin tone as a parameter. However, in response to your suggestion, we have decided to remove this variable from our analysis.
5 Methods – The chimeric antigen test has been applied for screening for chronic Chagas. While some basic description is helpful to the reader, likely some of the technical portion can be referenced to other papers.
Reply: We appreciate the reviewer's valuable suggestion. However, we kindly request that you reconsider your request. While we have previously described the technique in other articles, we believe it is beneficial to replicate it within this manuscript to ensure that readers do not need to refer to other sources. Therefore, we would like to maintain the methodological description of the immunoassays as it currently stands.
6.Results – not overstated. I assume among the 200 people living in the ZPS settlement the information on the 2 infected individuals is not identifiable to them specifically or that they gave permission for the details to be published? If not, I think the details could be made more general as to avoid identification.
Reply: The reviewer's point is valid, and we fully acknowledge the importance of ensuring the anonymity of research participants. It's important to note that both individuals who tested positive willingly consented to participate in the study, which gives us the confidence to share this information publicly.
7 Discussion- For the discussion, emphasize the findings of the study a lower rate of infection than expected. Explain why the findings are unique to the literature – lower than prior reports of Brazilian infection. I think discussing the use of the chimeric antigen screen as more specific would be helpful here and its use in other settings. Discuss the limitations of the study – small sample, but while not randomized it was a near complete survey of the community. Remove the details about the individuals here. I think the use of the chimeric antigens deserves a paragraph on the pros and cons. this may be a reason why the incidence was lower? Requiring transportation of samples and a full laboratory to process is a negative. How is cost comparison to running 2 conventional assays? The point of the paper may be that if you go out to do a seroprevalence study on Chagas, better to use a more specific test?
Reply: We greatly appreciate the reviewer's comments, which we have found to be pertinent. However, we kindly request the reviewer to reconsider the request. The primary objective of our study was to conduct a seroepidemiological survey encompassing the entire population residing in the Zumbi dos Palmares community. All aspects pertaining to the use of chimeric antigens in Chagas disease diagnosis have been extensively discussed in prior studies conducted by our research group. Therefore, introducing a new paragraph on this topic within the current study could divert the focus and shift the work towards the validation of the methodology. In previous research, we have established that the four antigens, owing to their composition, exhibit high sensitivity and specificity. They feature a broad range of repetitive and conserved epitopes from various T. cruzi proteins. We took care to exclude epitopes that displayed any similarity or homology with sequences of other pathogens from the composition of these antigens. However, should the reviewer deem this information genuinely relevant for enhancing the manuscript, we will certainly include it.
As per the editor's request, we have removed the details about the individuals as follows:
Before: The lack of positive cases in family members of the 22-year-old man born in Camamu with Chagas disease, suggests that congenital transmission was not the cause of his infection. Other possible routes of transmission, such as vector-borne transmission, should be considered, especially considering his migratory activities during the sugarcane and garlic harvests. Although we were unable to conduct clinical follow-up due to his temporary residence in another state (Espírito Santo) during our visits to the Zumbi dos Palmares settlement, he was referred to primary care in his municipality of residence. The other positive case, a 69-year-old man born in São Felipe, Bahia, was already aware of his arrhythmogenic cardiomyopathy caused by CD. This emphasizes the long-term health con-sequences and potential severity of the disease if left untreated.
After: The absence of positive cases among the family members of one individual infected with T. cruzi suggests that congenital transmission was not the cause of his infection. Other possible routes of transmission, such as vector-borne transmission, should be considered, especially considering his migratory activities during the sugarcane and garlic harvests. Although we were unable to conduct clinical follow-up due to his temporary residence in another state (Espírito Santo) during our visits to the Zumbi dos Palmares settlement, he was referred to primary care in his municipality of residence. The other individual infected with T. cruzi born in an endemic city of Bahia, was already diagnosed with arrhythmogenic cardiomyopathy attributable to CD. This emphasizes the long-term health con-sequences and potential severity of the disease if left untreated.
Best wishes,
Fred Santos (corresponding author)
Reviewer 3 Report
General Comments
The manuscript is well written and structured. The study is interesting, as it brings important information about a serological survey – a subject little studied but of extreme relevance in the context of public health. Various studies have been released in Bahia, indicating that some sites have higher circulation of T. cruzi in vectors. The proximity between infected vectors and humans is a clear indicator of the risk of infection in humans. In Rio Grande do Norte, several authors have indicated that high triatomine infestation densities and T. cruzi prevalence in domestic and peri-domestic habitats occur. On the other hand, In Paraíba, Ceará and Paraíba, only triatomines with low T. cruzi prevalence are found. How does this happen in the chosen study area? The authors talk about the possibility of vectorial transmission but do not even mention the possible triatomine species involved in the supposed transmission. Do these bugs are found in peri- domestic environments? I assume that the authors did ask the residents if they see bugs inside homes, but they should call those involved in vector control measures to rescue this information to mention as “pers. comm.”. To sum up, the work is important and deserves to be published, but these points need to be considered and considered.
Specific Comments
Introduction
Ln 56-66: “The World Health Organization estimates that 1.1 65 million people are infected with T. cruzi, and 25.4 million people are at risk of infection [9].”. Specify that it refers to worldwide infection because it continues a phase that talks about epidemiology in Brazil.
One point needs to be made clearer to the reader (in the introduction, but also in the method and discussion): the authors mention in the introduction that the acute phase of CD lasts a few weeks or months. It is necessary to explain whether the method can detect only these cases. Does the method detect chronic CD? This information should also be discussed.
Introduction and M&M: It is necessary to explain why this municipality was chosen (perhaps also in the discussion).
Discussion
Discuss the profile of housing structures in the studied area (even superficially). Are they suitable for triatomine infestation?
I recommend discussing (i) studies in the state of Bahia that deal with the eco-epidemiological profile, contemplating vectors. (ii) studies outside the state of Bahia where there is an intense circulation of T. cruzi. There are at least 3 studies mention that in Rio Grande do Norte the circulation of infected vectors close to humans was detected in a Chagasic outbreak in 2016-2017. I recommend connecting them with this study and with entomological surveys in the state of Bahia.
Author Response
How does this happen in the chosen study area? The authors talk about the possibility of vectorial transmission but do not even mention the possible triatomine species involved in the supposed transmission. Do these bugs are found in peri- domestic environments? I assume that the authors did ask the residents if they see bugs inside homes, but they should call those involved in vector control measures to rescue this information to mention as “pers. comm.”. To sum up, the work is important and deserves to be published, but these points need to be considered and considered.
Reply: This observation holds significant importance. Although the study site exhibits low endemicity for Chagas disease, it is a hotspot for other diseases such as HTLV, schistosomiasis, and various intestinal parasites, with strongyloidiasis being notably prevalent. Assessing Chagas disease positivity forms part of a comprehensive project aimed at diagnosing and treating multiple infections in individuals. Consequently, we carefully selected this specific population to extend the benefits beyond Chagas disease diagnosis, encompassing the detection and treatment of other infections. Our team queried residents about the presence of kissing bugs in and around their homes, but they reported no sightings. However, they have previously encountered these insects in other cities within the region, yet we couldn't identify the species based on their descriptions.
Specific Comments
Introduction. Ln 56-66: “The World Health Organization estimates that 1.1 65 million people are infected with T. cruzi, and 25.4 million people are at risk of infection [9].”. Specify that it refers to worldwide infection because it continues a phase that talks about epidemiology in Brazil.
Reply: We appreciate the reviewer for highlighting this matter. To enhance clarity, we have modified the sentence as follows:
Before: “…In Brazil, the country where T. cruzi was first identified [1], CD is a significant public health problem. The World Health Organization estimates that 1.1 million people are infected with T. cruzi, and 25.4 million people are at risk of infection [9]. However, the actual number of chronically infected individuals remains unknown, despite progress in interrupting CD transmission in Brazil.”
After: “…In Brazil, the country where T. cruzi was first identified [1], CD is a significant public health problem. Indeed, the World Health Organization estimates that 1.1 million people are infected with T. cruzi, and 25.4 million people are at risk of infection [9]. However, the actual number of chronically infected individuals remains unknown, despite progress in interrupting CD transmission in Brazil.
One point needs to be made clearer to the reader (in the introduction, but also in the method and discussion): the authors mention in the introduction that the acute phase of CD lasts a few weeks or months. It is necessary to explain whether the method can detect only these cases. Does the method detect chronic CD? This information should also be discussed.
Reply: We appreciate the reviewer's comment but kindly request a reconsideration of your suggestion. In the introduction, we explicitly state that our study focuses on chronic Chagas disease: (lines 74-77): “…Recognizing the challenges posed by CD, our study aimed to investigate the prevalence of chronic CD among individuals living in a rural community in Southern Bahia, representing an important step in understanding the disease burden in vulnerable populations.” Additionally, to enhance clarity, we have included the term 'chronic' in the discussion section.
Before: “…The overall prevalence of CD in the studied community was 0.92%, as detected by latent class analysis (LCA)…”
After: …The overall prevalence of chronic CD in the studied community was 0.92%, as detected by latent class analysis (LCA)…
Introduction and M&M: It is necessary to explain why this municipality was chosen (perhaps also in the discussion).
Reply: We appreciate the reviewer's request. To enhance the clarity of our text for readers, we have included the following sentence in the discussion:
Before: “Chagas disease (CD) continues to be a significant public health issue in Brazil, with a high burden of undiagnosed cases [21]. In this study, we conducted a seroepidemiological survey to determine the prevalence of CD in an impoverished rural community in Southern Bahia. Our findings highlight the importance of active surveillance and early detection of CD in vulnerable populations. The overall prevalence of chronic CD in the studied community was 0.92%, as detected by latent class analysis (LCA). This prevalence is relatively low compared to the estimated national prevalence of CD in Brazil [24]. However, it is important to note that even in areas of low prevalence, identifying positive cases through seroepidemiological surveys is crucial for timely diagnosis and management.”
After: Chagas disease (CD) continues to be a significant public health issue in Brazil, with a high burden of undiagnosed cases [21]. In this study, we conducted a seroepidemiological survey to determine the prevalence of CD in an impoverished rural community in Southern Bahia. While the study site falls under a low endemicity category for CD, it is endemic for other infections, including HTLV, schistosomiasis, and intestinal parasites, with strongyloidiasis being notably prevalent. The evaluation of CD positivity is part of a broader project focused on diagnosing and treating individuals for multiple infections. Our findings highlight the importance of active surveillance and early detection of CD in vulnerable populations. The overall prevalence of chronic CD in the studied community was 0.92%, as detected by latent class analysis (LCA). This prevalence is relatively low compared to the estimated national prevalence of CD in Brazil [24]. However, it is important to note that even in areas of low prevalence, identifying positive cases through seroepidemiological surveys is crucial for timely diagnosis and management.
Discussion. Discuss the profile of housing structures in the studied area (even superficially). Are they suitable for triatomine infestation?
Reply: Thank you for bringing this to our attention. We have added a sentence to the discussion section to provide a more comprehensive characterization of the local epidemiological scenario:
Before: “The prevalence of CD in the studied rural community was lower than expected, considering the socioeconomic conditions and environmental factors that can contribute to T. cruzi transmission. A systematic review and me-ta-analysis published in 2014 reported a mean prevalence of 4.2% for Brazil…”
After: The prevalence of CD in the studied rural community was lower than expected, considering the socioeconomic conditions and environmental factors that can contribute to T. cruzi transmission. Although present, we observed few mud houses with cracks in the area, and the residents reported not encountering the bugs. A systematic review and me-ta-analysis published in 2014 reported a mean prevalence of 4.2% for Brazil…
I recommend discussing (i) studies in the state of Bahia that deal with the eco-epidemiological profile, contemplating vectors. (ii) studies outside the state of Bahia where there is an intense circulation of T. cruzi. There are at least 3 studies mention that in Rio Grande do Norte the circulation of infected vectors close to humans was detected in a Chagasic outbreak in 2016-2017. I recommend connecting them with this study and with entomological surveys in the state of Bahia.
Reply: We sincerely thank the reviewer for the suggestion but kindly request a reconsideration of the request. While the suggested data is related, its inclusion would extend beyond the scope of our active case search perspective.
Round 2
Reviewer 2 Report
The corrections were helpful . The research is appropriate for publication.
Reviewer 3 Report
General Comments
The authors have done a good job in reviewing the manuscript, considering the recommendations and suggestions whenever they found them suitable. Therefore, in my opinion, the manuscript can now be accepted by the journal.